# BAIL: Best-Action Imitation Learning for Batch Deep Reinforcement Learning

## Abstract

The field of Deep Reinforcement Learning (DRL) has recently seen a surge in research in batch reinforcement learning, which aims for sample-efficient learning from a given data set without additional interactions with the environment. In the batch DRL setting, commonly employed off-policy DRL algorithms can perform poorly and sometimes even fail to learn altogether. In this paper we propose a new algorithm, Best-Action Imitation Learning (BAIL), which unlike many off-policy DRL algorithms does not involve maximizing Q functions over the action space. Striving for simplicity as well as performance, BAIL first selects from the batch the actions it believes to be high-performing actions for their corresponding states; it then uses those state-action pairs to train a policy network using imitation learning. Although BAIL is simple, we demonstrate that BAIL achieves state of the art performance on the Mujoco benchmark.

## 1 Introduction

The field of Deep Reinforcement Learning (DRL) has recently seen a surge in research in batch reinforcement learning, which is the problem of sample-efficient learning from a given data set without additional interactions with the environment. Batch reinforcement learning is appealing because it dis-entangles policy optimization (exploitation) from data collection (exploration). This enables reusing the data collected by a policy to possibly improve the policy without further interactions with the environment. Furthermore, a batch learning reinforcement learning algorithm can potentially be deployed as part of a growing-batch algorithm, where the batch algorithm seeks a high-performing exploitation policy using the data in an experience replay buffer, combines this policy with exploration to add fresh data to the buffer, and then repeats the whole process (Lange et al., 2012).

Fujimoto et al. (2018a) recently made the critical observation that commonly employed off-policy algorithms based on Deep Q-Learning (DQL) often perform poorly and sometimes even fail to learn altogether. Indeed, off-policy DRL algorithms typically involve maximizing an approximate Q-function over the action space (Lillicrap et al., 2015; Fujimoto et al., 2018b; Haarnoja et al., 2018a), leading to an extrapolation error, particularly for state-action pairs that are not in the batch distribution. Batch-Constrained deep Q-learning (BCQ), which obtains good performance for many of the Mujoco environments (Todorov et al., 2012), avoids the extrapolation error problem by constraining the set of actions over which the approximate Q-function is optimized (Fujimoto et al., 2018a).

We propose a new algorithm, Best-Action Imitation Learning (BAIL), which strives for both simplicity and performance. BAIL does not suffer from the extrapolation error problem since it does not maximize over the action space in any step of the algorithm. BAIL is simple, thereby satisfying the principle of Occam's razor.

The BAIL algorithm has two steps. In the first step, it selects from the batch a subset of state-action pairs for which the actions are believed to be good actions for their corresponding states. In the second step, it simply trains a policy network with imitation learning using the selected actions from the first step. To find the best actions, we train a neural network to obtain the "upper envelope" of the Monte Carlo returns in the batch data, and then we select from the batch the state-action pairs that are near the upper envelope. We believe the concept of the upper-envelope of a data set is also novel and interesting in its own right.

Because the BCQ code is publicly available, we are able to make a careful comparison of the performance of BAIL and BCQ. We do this for batches generated by training DDPG (Lillicrap et al., 2015) for the Half-Cheetah, Walker, and Hopper environments, and for batches generated by training Soft Actor Critic (SAC) for the Ant environment (Haarnoja et al., 2018a;b). Although BAIL is simple, we demonstrate that BAIL achieves state of the art performance on the Mujoco benchmark, often outperforming Batch Constrained deep Q-Learning (BCQ) by a wide-margin. We also provide anonymized code for reproducibility[1].

## 2   RELATED WORK

Batch reinforcement learning in both the tabular and functional approximator settings has long been studied (Lange et al., 2012; Strehl et al., 2010) and continues to be a highly active area of research (Swaminathan & Joachims, 2015; Jiang & Li, 2015; Thomas & Brunskill, 2016; Farajtabar et al., 2018; Irpan et al., 2019; Jaques et al., 2019). Imitation learning is also a well-studied problem (Schaal, 1999; Argall et al., 2009; Hussein et al., 2017) and also continues to be a highly active area of research (Kim et al., 2013; Piot et al., 2014; Chemali & Lazaric, 2015; Hester et al., 2018; Ho et al., 2016; Sun et al., 2017; 2018; Cheng et al., 2018; Gao et al., 2018).

This paper relates most closely to (Fujimoto et al., 2018a), which made the critical observation that when conventional DQL-based algorithms are employed for batch reinforcement learning, performance can be very poor, with the algorithm possibly not learning at all. Off-policy DRL algorithms involve maximizing an approximate action-value function $Q(s, a)$ over all actions in the action space. (Or over the actions in the manifold of the parameterized policy.) The approximate action-value function can be very inaccurate, particularly for state-action pairs that are not in the state-action distribution of the batch (Fujimoto et al., 2018a). Due to this extrapolation error, poor-performing actions can be chosen when optimizing $Q(s, a)$ over all actions. With traditional off-policy DRL algorithms (such as DDPG (Lillicrap et al., 2015), TD3 (Fujimoto et al., 2018b) and SAC (Haarnoja et al., 2018a)), if the action-value function over-estimates a state-action pair, the policy will subsequently collect new data in the over-estimated region, and the estimate will get corrected. In the batch setting, however, where there is no further interaction with the environment, the extrapolation error is not corrected, and the poor choice of action persists in the policy (Fujimoto et al., 2018a).

Batch-Constrained deep Q-learning (BCQ) avoids the extrapolation error problem by constraining the set of actions over which the approximate Q-function is optimized (Fujimoto et al., 2018a). More specifically, BCQ first trains a state-dependent Variational Auto Encoder (VAE) using the state action pairs in the batch data. When optimizing the approximate Q-function over actions, instead of optimizing over all actions, it optimizes over a subset of actions generated by the VAE. The BCQ algorithm is further complicated by introducing a perturbation model, which employs an additional neural network that outputs an adjustment to an action. BCQ additionally employs a modified version of clipped-Double Q-Learning to obtain satisfactory performance. We show experimentally that our much simpler BAIL algorithm typically performs better than BCQ by a wide margin.

Kumar et al. (2019) recently proposed BEAR for batch DRL. BEAR is also complex, employing Maximum Mean Discrepancy (Gretton et al., 2012), kernel selection, a parametric model that fits a tanh-Gaussian distribution, and a test policy that is different from the learned actor policy. In this paper we do not experimentally compare BAIL with BEAR since the code for BEAR is not publicly available at the time of writing.

Agarwal et al. (2019) recently proposed another algorithm for batch DRL called Random Ensemble Mixture (REM), an ensembling scheme which enforces optimal Bellman consistency on random convex combinations of the Q-heads of a multi-headed Q-network. For the Atari 2600 games, batch REM can out-perform the policies used to collect the data. REM and BAIL are orthogonal, and it may be possible to combine them in the future to achieve even higher performance. No experimental results are provided for REM applied to the Mujoco benchmark (Agarwal et al., 2019).

---

[1]https://anonymous.4open.science/r/e5fbe703-a32d-4679-a2a8-095e74b96e85

## 3   BATCH DEEP REINFORCEMENT LEARNING

We represent the environment with a Markov Decision Process (MDP) defined by a tuple $(\mathcal{S}, \mathcal{A}, g, r, \rho, \gamma)$, where $\mathcal{S}$ is the state space, $\mathcal{A}$ is the action space, $\rho$ is the initial state distribution, and $\gamma$ is the discount factor. The functions $g(s, a)$ and $r(s, a)$ represent the dynamics and reward function, respectively. In this paper we assume that the dynamics of the environment are deterministic, that is, there are real-valued functions $g(s, a)$ and $r(s, a)$ such that when in state $s$ and action $a$ is chosen, then the next state is $s' = g(s, a)$ and the reward received is $r(s, a)$. We note that all the simulated robotic locomotion environments in the Mujoco benchmark are deterministic, and many robotic tasks are expected to be deterministic environments. Furthermore, many of the Atari game environments are deterministic (Bellemare et al., 2013). Thus, from an applications perspective, the class of deterministic environments is a large and important class. Although we assume that the environment is deterministic, as is typically the case with reinforcement learning, we do not assume the functions $g(s, a)$ and $r(s, a)$ are known.

In batch reinforcement learning, we are provided a batch of $m$ data points $\mathcal{B} = \{(s_i, a_i, r_i, s'_i), \ i = 1, ..., m\}$. We assume $\mathcal{B}$ is fixed and given, and there is no further interaction with the environment. Often the batch $\mathcal{B}$ is training data, generated in some episodic fashion. However, in the batch reinforcement learning problem, we do not have knowledge of the algorithm, models, or seeds that were used to generate the episodes in the batch $\mathcal{B}$.

Typically the batch data is generated during training with a non-stationary DRL policy. After training, the original DRL algorithm produces a *final-DRL policy*, with exploration turned off. In our numerical experiments, we will compare the performance of policies obtained by batch algorithms with the performance of the final-DRL policy. Ideally, we would like the performance of the batch-derived policy to be as good or better than the final-DRL policy.

The case where batch data is generated from a non-stationary training policy is of particular interest because it is typically a rich data set from which it may be possible to derive high-performing policies. Furthermore, a batch learning algorithm can potentially be deployed as part of a growing-batch algorithm, where the batch algorithm seeks a high-performing exploitation policy using the current data in an experience replay buffer, combines this policy with exploration to add fresh data to the buffer, and then repeats the whole process (Lange et al., 2012).

## 4   BEST-ACTION IMITATION LEARNING (BAIL)

In this paper we present BAIL, an algorithm that not only performs well on simulated robotic locomotion tasks, but is also conceptually and algorithmically simple. BAIL has two steps. In the first step, it selects from the batch data $\mathcal{B}$ the state-action pairs for which the actions are believed to be good actions for their corresponding states. In the second step, we simply train a policy network with imitation learning using the selected actions from the first step.

Many approaches could be employed to select the best actions. In this paper we propose training a single neural network to create an *upper envelope* of the Monte Carlo returns, and then selecting the state-action pairs in the batch $\mathcal{B}$ that have returns near the upper envelope.

### 4.1   UPPER ENVELOPE

We first define a $\lambda$-smooth upper envelope, and then provide an algorithm for finding it. To the best of our knowledge, the notion of the upper envelope of a data set is novel.

Recall that we have a batch of data $\mathcal{B} = \{(s_i, a_i, r_i, s'_i), \ i = 1, ..., m\}$. Although we do not assume we know what algorithm was used to generate the batch, we make the natural assumption that the data in the batch was generated in an episodic fashion, and that the data in the batch is ordered accordingly. For each data point $i \in \{1, \ldots, m\}$, we calculate an approximate Monte Carlo return $G_i$ as the sum of the discounted returns from state $s_i$ to the end of the episode:

$$G_i = \sum_{t=i}^{T_i} \gamma^{t-i} r_t \tag{1}$$

where $T_i$ denotes the time at which the episode ends for the data point $s_i$. The Mujoco environments are naturally infinite-horizon non-episodic continuing-task environments (Sutton & Barto, 2018). During training, however, researchers typically create artificial episodes of length 1000 time steps; after 1000 time steps, a random initial state is chosen and a new episode begins. Because the Mujoco environments are continuing tasks, it is desirable to approximate the return over the infinite horizon, particularly for $i$ values that are close to the (artificial) end of an episode. To do this, we note that the data-generation policy from one episode to the next typically changes slowly. We therefore apply a simple augmentation heuristic of concatenating the subsequent episode to the current episode, and running the sum in (1) to infinity. (In practice, we end the sum when the discounting reduces the contribution of the rewards to a negligible amount.) Our ablation study in the Appendix shows that this simple heuristic can significantly improve performance. Note this approach also obviates the need for knowing when new episodes begin in the data set $\mathcal{B}$.

Having defined the return for each data point in the batch, we now seek an upper-envelope $V(s)$ for the data $\mathcal{G} := \{(s_i, G_i), \ i = 1, ..., m\}$. Let $V_\phi(s)$ be a neural network with parameters $\phi$ that takes as input a state $s$ and outputs a real number. We say that $V_{\phi^*}(s)$ is a $\lambda$-smooth upper envelope for $\mathcal{G}$ if it has the following properties:

1. $V_{\phi^*}(s_i) \geq G_i$ for all $i = 1, \ldots, m$.
2. Among all the parameterized functions $V_\phi(s)$ satisfying condition 1, it minimizes:

$$L(\phi) = \sum_{i=1}^{m} [V_\phi(s_i) - G_i]^2 + \lambda \|\phi\|^2 \tag{2}$$

where $\lambda$ is a non-negative constant. An upper-envelope is thus a smooth function that lies above all the data points, but is nevertheless close to the data points.

**Theorem 4.1.** *Suppose that $V_{\phi^*}(s)$ is a $\lambda$-smooth upper envelope for $\mathcal{G}$. Then,*

*(1) $V_{\phi^*}(s) = \max\{G_i : i = 1, 2, \ldots, m\}$ as $\lambda \to \infty$.*

*(2) If there is sufficient capacity in the network and $\lambda = 0$, then the $V_{\phi^*}$ interpolates the data in $\mathcal{G}$. For example, if $\lambda = 0$ and $V_\phi(s)$ is a neural network with ReLU activation functions with at least $2m + d$ weights and two layers, where $d$ is the dimension of the state space $\mathcal{S}$, then $V_{\phi^*}(s_i) = G_i$ for all $i = 1, 2, \ldots, m$.*

From the above theorem, we see that when $\lambda$ is very small, the upper envelope aims to interpolate the data, and when $\lambda$ is large, the upper envelope approaches a constant going through the highest data point. Just as in classical regression, there is a sweet-spot for $\lambda$, the one that provides the best generalization.

We note that there are other natural approaches for defining an upper-envelope, some based on alternative loss functions, others based on data clustering without making use of function approximators. Also, it may be possible to combine episodes to generate improved upper envelopes. These are all questions for future research.

To obtain an approximate upper envelope of the data $\mathcal{G}$, we employ classic regression with a modified loss function, namely,

$$L(\phi) = \sum_{i=1}^{m} (V_\phi(s_i) - G_i)^2 \{ \mathbb{1}_{(V_\phi(s_i) > G_i)} + K \mathbb{1}_{(V_\phi(s_i) < G_i)} \} + \lambda \|\phi\|^2 \tag{3}$$

where $K >> 1$ and $\mathbb{1}_{(\cdot)}$ is the indicator function.

For a finite $K$ value, the above loss function will only produce an approximate upper envelope, since it is possible $V(s_i)$ may be slightly less than $G_i$ for a few data points. In practice, we find $K = 10,000$ works well for all environments tested. When $K \to \infty$, the approximation becomes exact, as stated in the following:

**Theorem 4.2.** *Let $\phi^*$ be an optimal solution that minimizes $L(\phi)$. Then, when $K \to \infty$, $V_{\phi^*}(s)$ will be an exact $\lambda$-smooth upper envelope.*

Also, instead of L2 regularization, in practice we employ the simpler early-stopping regularization, thereby obviating a search for the parameter $\lambda$. We also clip the upper envelope at values near $\max_i G_i$, as described in the appendix, which can potentially provide further gains in performance.

### 4.2 Selecting the best actions

The BAIL algorithm employs the upper envelope to select the best $(s, a)$ pairs from the batch data $\mathcal{B}$. It then uses ordinary imitation learning (behavioral cloning) to train a policy network using the selected actions. Let $V(s)$ denote the upper envelope obtained from minimizing the loss function (3) for a fixed value of $K$.

We consider two approaches for selecting the best actions. In the first approach, which we call BAIL-border, we choose all $(s_i, a_i)$ pairs from the batch data set $\mathcal{B}$ such that

$$G_i > xV(s_i) \tag{4}$$

We set $x$ such that $p\%$ of the data points are selected, where $p$ is a hyper-parameter. In this paper we use $p = 25$ for all environments and batches. Thus BAIL-border chooses state-action pairs whose returns are near the upper envelope. The pseudo-code for the Bail-border algorithm is given in the appendix.

In the second approach, which we refer to as BAIL-TD, we select a pair $(s_i, a_i)$ if

$$r_i + \gamma V(s_i') > xV(s_i) \tag{5}$$

where $x$ is a hyper-parameter close to 1. Thus BAIL-TD chooses state-action pairs for which backed-up estimated return $r_i + \gamma V(s_i')$ is close to the upper envelope value $V(s_i)$.

In summary, BAIL employs two neural networks. The first neural network is used to approximate a value function based on the data in the batch $\mathcal{B}$. The second neural network is the policy network, which is trained with imitation learning. This simple approach does not suffer from extrapolation error since it does not perform any optimization over the action space. An algorithmic description of BAIL is given in Algorithm 1.

---

**Algorithm 1** BAIL

1: Initialize upper envelope parameters $\phi$, policy parameter $\theta$, obtain batch data $\mathcal{B}$.
2: Compute return for each data point $i$: $G_i = \sum_{t=i}^{T_i} \gamma^{t-i} r_t$
3: Obtain upper envelope by minimizing the loss:
4: **for** $j = 1, \ldots, J$ **do**
5:     sample a minibatch of data B from the batch $\mathcal{B}$
6:     minimize over $\phi$: $L(\phi) = \sum_{i=1}^{|B|} (V_\phi(s_i) - G_i)^2 \{\mathbb{1}_{(V_\phi(s_i)>G_i)} + K\mathbb{1}_{(V_\phi(s_i)<G_i)}\} + \lambda\|\phi\|^2$
7: Select data $i$ for $G_i > xV_\phi(s_i)$, select $x$ so that $p\%$ data in $\mathcal{B}$ are selected, let $\mathcal{U}$ be set of selected data
8: **for** $l = 1, \ldots, L$ **do**
9:     sample a minibatch $U$ of data from $\mathcal{U}$
10:     minimize over $\theta$: $L(\theta) = \sum_{i=1}^{|U|} (\pi_\theta(s_i) - a_i)^2$

---

## 5 Experimental results

We carried out experiments with four of the most challenging environments in the Mujoco benchmark (Todorov et al., 2012) of OpenAI Gym. For the environments Hopper-v2, Walker-v2 and HalfCheetah-v2, we used the "Final Buffer" batch exactly as described in Fujimoto et al. (2018a) to allow for a fair comparison with BCQ. Specifically, we trained DDPG for one million time steps with $\sigma = 0.5$ to generate a batch. For the environment Ant-v2, we trained adaptive SAC (Haarnoja et al., 2018b) again for one million time steps to generate a batch.

In our experiments, we found that different batches generated with different seeds but with the same algorithm in the same environment can lead to surprisingly different results for batch DRL algorithms. To address this, for each of the environments we generated four batches, giving a total of 16 data sets.

Figure 1 provides visualizations of 4 of the 16 upper envelopes, one for each of the 4 environments. In each visualization, the data points in the corresponding batch are ordered according to their upper-envelope values $V(s_i)$. With this new ordering, the figure plots $(s_i, G_i)$ for each of the one million

data points. The monotonically increasing blue line is the the upper envelope obtained by minimizing $V(s)$. Note that in all of the figures, a small fraction of the data points are above upper envelopes due to the finite value of $K = 10,000$. But also note that the upper envelope mostly hugs the data. The constant black line is the clipping value. The final upper envelope is the minimum of the blue and black lines. All 16 upper envelopes are shown in the appendix.

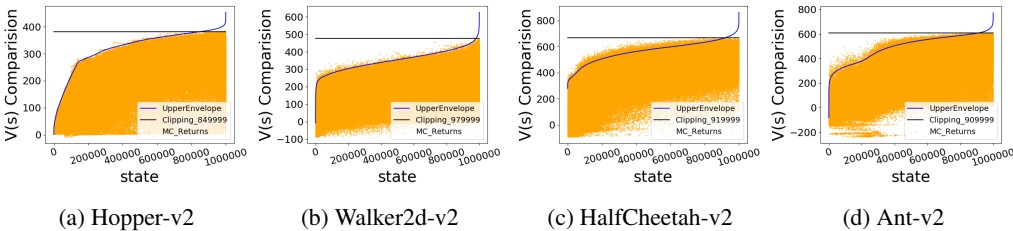

| (a) Hopper-v2 | (b) Walker2d-v2 | (c) HalfCheetah-v2 | (d) Ant-v2 |

Figure 1: Five illustrative upper envelopes trained from data with adaptive clipping

Figure 2 compares the performance of BAIL, BCQ, Behavioral Cloning (BC), and the final DDPG/SAC policy for the four environments. When training with BCQ, we used the code provided by the authors (Fujimoto et al., 2018a). Because at the time of writing the code for BEAR was not available, we do not compare our results with BEAR (Kumar et al., 2019). Also, all the results presented in this section are for BAIL-border, which we simply refer to as BAIL. In the appendix we provide results for BAIL-TD. The x-axis is the number of parameter updates and the y-axis is the test return averaged over 10 episodes. BAIL, BCQ, and BC are each trained with five seeds. The figure shows the mean and standard deviation confidence intervals for these values. The figure also shows test result of the final-DDPG/SAC policy. This value is obtained by averaging test results from the last 100,000 timesteps (of one million time steps). During this period, test performance of SAC and particularly DDPG can greatly fluctuate with relatively small improvement on average. We calculate the mean and standard deviation of the test results over this period, plot the mean as a straight line, and use the transparent green background to show the confidence intervals. This enables us to fairly compare the performance of the final test policy obtained with the behavioral algorithm with the test policies from the batch algorithms.

We make the following observations. For Hopper, Walker and Ant, BAIL always beats BCQ usually by a wide margin. For HalfCheetah, BAIL does better than BCQ for half of the batches. In almost all of the curves, BAIL has a much lower confidence interval than BCQ. Perhaps more importantly, BAIL's performance is stable over training, whereas BCQ can vary dramatically. (This is a serious issue for batch reinforcement learning, since it cannot interact with the environment to find the best stopping point.) Importantly, BAIL also performs as well or better than the Final-DDPG/SAC policy in all but of the 16 batches. This gives promise that BAIL, or a future variation of BAIL, could also be employed within a growing-batch algorithm.

Table 1: Performance comparison at one million samples (mean and std over batches and random seeds). Last column shows percentage improvement of BAIL over BCQ.

| Environment | Final-DDPG/SAC | BCQ | BAIL | Improvement |
|---|---|---|---|---|
| Hopper-v2 | **2547.7** ± 750.4 | 1468.9 ± 552.6 | 2437.5 ± 489.7 | 65.9% |
| Walker2d-v2 | 1742.1 ± 656.3 | 2020.2 ± 699.3 | **2496.7** ± 409.9 | 23.6% |
| HalfCheetah-v2 | 2612.4 ± 342.2 | 2449.7 ± 267.7 | **2660.0** ± 77.7 | 8.6% |
| Ant-v2 | 4506.0 ± 483.6 | 4315.6 ± 416.4 | **4630.7** ± 310.9 | 7.3% |

We also summarize the results in Table 1. For this table, we average the performance of each algorithm over four batches, using the performance values at one million updates. Table 1 shows that BAIL's performance is better than that of BCQ for all four environments, with a 66% and 23% average improvement for Hopper and Walker, respectively. BAIL also beats the Final-DDPG/SAC policies in three of the environments, and has significantly lower variance.

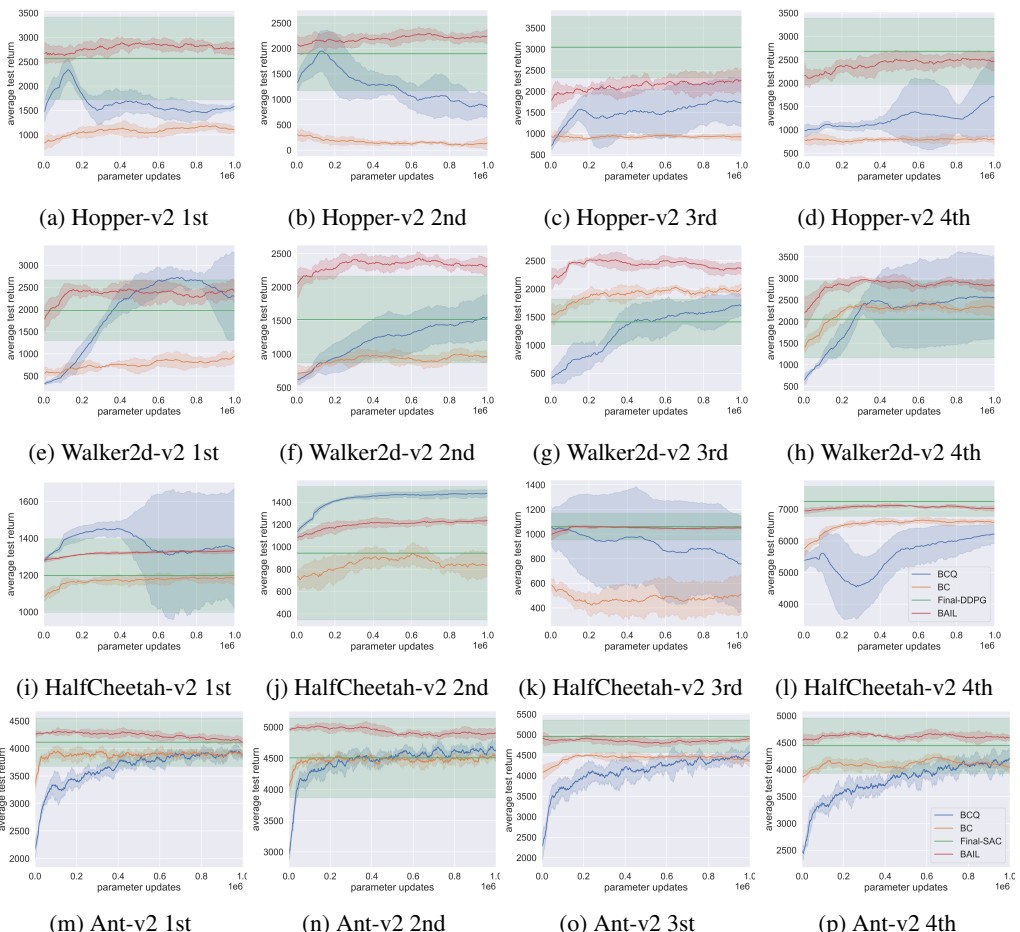

Figure 2: Performance comparison of BAIL, BCQ, BC and Final-DDPG/SAC

In the appendix we also provide experimental results for DDPG batches generated with $\sigma = 0.1$, which is similar to the "Concurrent" dataset in Fujimoto et al. (2018a). For this low noise level 0.1, BAIL continues to beat BCQ by a wide margin for Hopper and Walker, and continues to beat Final-DDPG for half of the batches. However, in the low noise case for HalfCheetah, BCQ beats BAIL for 3 of the 4 batches.

## 5.1 Ablation Study

BAIL uses an upper envelope to select the "best" data points for training a policy network with imitation learning. We have shown that BAIL typcially beats ordinary behavioral cloning and BCQ by a wide margin, and often performs better than the Final-DDPG and Final-SAC policies. But it is natural to ask how BAIL performs when using more naive approaches for selecting the best actions. We consider two naive approaches. The first approach, "Highest Returns," is to select from the batch the 25% of data points that have the highest $G_i$ values. The second approach, "Recent Data," is to select the last 25% data points from the batch. Figure (3) shows the results for all four environments. We see that for each environment, the upper envelope approach is the winner for most of the batches: for Hopper, the upper envelope wins for all four batches by a wide margin; for Walker the upper-envelope approach wins by a wide margin for two batches, and ties Highest Returns for two batches; for HalfCheetah, the upper-envelope approach wins for three batches and ties Highest Returns for one batch; and for Ant, the upper-envelope approach wins for three batches and ties the other two approaches for the other batch. We conclude, for the environments and batch generation mechanisms considered in this paper, the upper envelope approach performs significantly better and is more robust than both naive approaches.

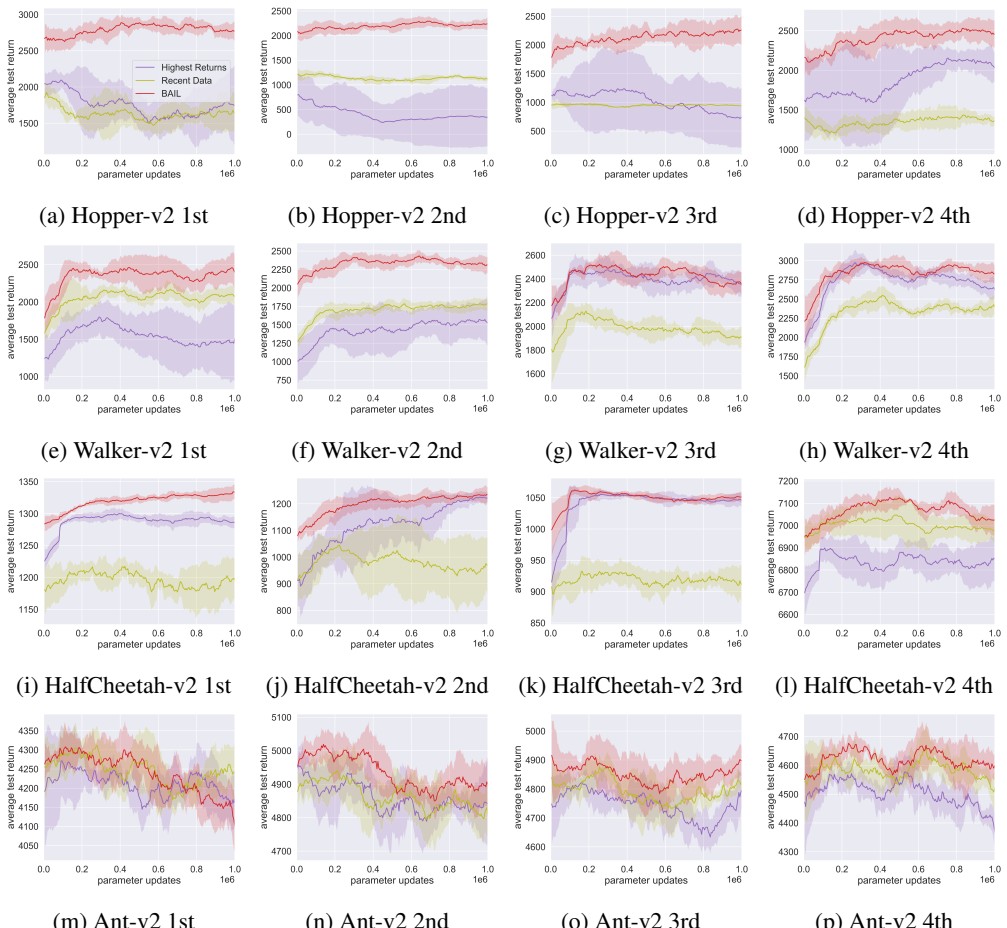

Figure 3: Comparison of BAIL scheme with Highest Returns and with Recent Samples schemes. All schemes use 25% of the data in the batch.

In the Appendix we provide additional ablation studies. Our experimental results show that modifying the returns to approximate infinite horizon returns is often useful for BAIL's performance, and that clipping the upper envelope also provides gains although much more modest.

In summary, our experimental results show that BAIL achieves state-of-the-art performance, and often beats BCQ by a wide margin. Moreover, BAIL's performance is stable over training, whereas BCQ typically varies dramatically over training. Finally, BAIL achieves this superior performance with an algorithm that is much simpler than BCQ.

## 6 CONCLUSION

Although BAIL as described in this paper is simple and gives state-of-the-art performance, there are several directions that could be explored in the future for extending BAIL. One avenue is generating multiple upper envelopes from the same batch, and then ensembling or using a heuristic to pick the upper envelope which we believe would give the best performance. A second avenue is to optimize the policy by modifying the best actions. A third avenue is to assign weights to the state-action pairs when training with imitation learning. And a fourth avenue is to explore designing a growing batch algorithm which uses BAIL as a subroutine for finding a high-performing exploitation policy for each batch iteration.

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

## A PROOFS

### A. Proof of Theorem 4.1

*Proof.* First, let us consider the case when $\lambda \to +\infty$. We can re-write the definition of the upper envelope as a constrained optimization problem:

$$\min_{\phi} \sum_{i=1}^{m}[V_{\phi}(s_i) - G_i]^2 + \lambda\|\phi\|^2 \tag{6}$$

$$s.t. \qquad G_i - V_{\phi}(s_i) \leq 0, \qquad i = 1, 2 \ldots, m$$

where $V_{\phi^*}$ is the optimal solution to the above optimization problem. We write the Lagrangian function:

$$L(\phi, \mu) = \sum_{i=1}^{m}[V_{\phi}(s_i) - G_i]^2 + \lambda\|\phi\|^2 + \sum_{i=1}^{m}\mu_i[G_i - V_{\phi}(s_i)] \tag{7}$$

As the optimal solution, $V_{\phi^*}$ must satisfy the KKT conditions specified below:

$$\begin{cases} \frac{\mathrm{d}L(\phi,\mu)}{\mathrm{d}\phi}|_{\phi=\phi^*} = 0 \\ \mu_i[G_i - V_{\phi^*}(s_i)] = 0, & i = 1, 2, \ldots, m \\ \mu_i \geq 0, & i = 1, 2, \ldots, m \end{cases} \tag{8}$$

Suppose $\phi = (\phi_1, \ldots, \phi_n)$, by the first KKT condition, we have

$$\frac{\partial}{\partial\phi_j}\sum_{i=1}^{m}[V_{\phi}(s_i) - G_i]^2|_{\phi_j=\phi_j^*} + 2\lambda\phi_j^* + \frac{\partial}{\partial\phi_j}\sum_{i=1}^{m}\mu_i[G_i - V_{\phi}(s_i)]|_{\phi_j=\phi_j^*} = 0, \qquad j = 1, 2, \ldots, n.$$

So we have:

$$\phi_j^* = -\frac{1}{2\lambda}\frac{\partial}{\partial\phi_j}\sum_{i=1}^{m}[V_{\phi}(s_i) - G_i]^2|_{\phi_j=\phi_j^*} - \frac{1}{2\lambda}\frac{\partial}{\partial\phi_j}\sum_{i=1}^{m}\mu_i[G_i - V_{\phi}(s_i)]|_{\phi_j=\phi_j^*}, \qquad j = 1, 2, \ldots, n.$$

When $\lambda \to \infty$, we have $\phi^* = 0$. In this case, it follows that $V_{\phi^*} = C$ for some constant $C$. As $V_{\phi^*}(s_i) \geq G_i$, in order to minimize (3) we must have $C = \max\{G_i, \ i = 1, 2, \ldots, m\}$.

For the case of $\lambda = 0$, notice that we only have finitely many input $s_i$ to be the input of the neural network. Therefore, this is a typical problem regarding the *finite-sample expressivity* of the neural networks, and the proof directly follows from the work done by Zhang et al. (Zhang et al., 2017). $\square$

### B. Proof of Theorem 4.2

*Proof.* Let $\phi^*$ be the optimal value that minimizes $L(\phi)$. Let's proceed by contradiction and assume that there exists some $k$ such that

$$V_{\phi^*}(s_k) < G_k$$

Let $\phi'$ be an arbitrary given value such that $V_{\phi'}(s_i) \geq G_i$ for all $i \in \{1, 2, \ldots, m\}$. Then we have

$$\sum_{i=1}^{m}(V_{\phi^*}(s_i) - G_i)^2 \mathbb{1}_{(V_{\phi^*}(s_i)>G_i)} + K(V_{\phi^*}(s_k) - G_k)^2 + \lambda\|\phi^*\|^2 \leq L(\phi^*)$$

$$L(\phi^*) \leq L(\phi') = \sum_{i=1}^{m}(V_{\phi'}(s_i) - G_i)^2 + \|\phi'\|^2$$

This implies that

$$\sum_{i=1}^{m}(V_{\phi^*}(s_i) - G_i)^2 \mathbb{1}_{(V_{\phi^*}(s_i)>G_i)} + K(V_{\phi^*}(s_k) - G_k)^2 + \lambda\|\phi^*\|^2 \leq \sum_{i=1}^{m}(V_{\phi'}(s_i) - G_i)^2 + \|\phi'\|^2$$

which is impossible when $K \to \infty$. Therefore, we must have $V_{\phi^*}(s_i) \geq G_i$ for all $i \in \{1, 2, \ldots, m\}$ as $K \to \infty$. In this way, when $K \to \infty$, $\phi^*$ actually minimizes

$$L(\phi) = \sum_{i=1}^{m}(V_{\phi}(s_i) - G_i)^2 + \|\phi\|^2$$

which completes the proof. $\square$

# B    IMPLEMENTATION DETAILS AND HYPERPARAMETERS

## B.1    IMPLEMENTATION OF BAIL ALGORITHM

Table 2: Upper Envelope Hyperparameters

| Parameter | Value |
|---|---|
| optimizer | Adam (Kingma & Ba, 2014) |
| learning rate | $3 \cdot 10^{-3}$ |
| discount ($\gamma$) | 0.99 |
| regularization constant $\lambda$ | $2 \cdot 10^{-2}$ |
| $K$ | 10,000 |
| number of hidden units | $128 \times 128$ |

Table 3: BAIL Hyperparameters

| Parameter | Value |
|---|---|
| data in batch | $10^6$ |
| optimizer | Adam (Kingma & Ba, 2014) |
| learning rate | $10^{-3}$ |
| regularization constant $\lambda$ | 0 |
| mini-batch size | 100 |
| BAIL-border $p\%$ | 25% |
| BAIL-TD $x$ | 0.96 |
| number of hidden units | $400 \times 300$ |

## B.2    IMPLEMENTATION OF COMPETING ALGORITHMS

For the behavioral DDPG algorithm, we used the implementation of Fujimoto et al. (2018b). For the behavioral SAC algorithm, we implemented it in Pytorch, mainly following the pseudocode provided by (Achiam), and used hyperparameters in Haarnoja et al. (2018b). For the BCQ algorithm, we used the authors' implementation (Fujimoto et al., 2018a). For behavioral cloning and its variants in the ablation study section, the network structure, learning rate, mini-batch size, and so on are identical to those in Table 3 for BAIL.

For the upper envelope network, our network has two hidden layers as does the Q network in BCQ and SAC. However, the number of hidden units in our network is less than those used in BCQ ($400 \times 300$) and SAC ($256 \times 256$). In future work we will see if we can obtain further improvements with BAIL using larger networks for the upper envelope.

## B.3    CLIPPING HEURISTICS

As is shown in Figure 1, in practice the trained upper envelope does not always fit well the data points on the right side of the plots, where the upper envelope can become very large. In that region, the data points with the highest returns will not be selected as "best actions" and therefore not used for imitation learning step. We observe that if we plot the upper envelope values for all the states in the buffer in ascending order as is shown in Figure 1, the upper envelope value $V(s_i)$ starts to deviate from the Monte Carlo return $G_i$ at a point where $V(s_i) \approx \max\{G_i\}$. We therefore use the following heuristic. We say that the upper envelope value begins to deviate from the Monte Carlo return at state $s_i$ if $V(s_j) > G_j$ for $i \leq j < i + 10000$. We set the clipping value $C = V(s')$ where $s'$ is the starting point of this deviation. Then the actual UE values used to select data is $\min\{V(s), C\}$. In practice, the clipping heuristic gives a small boost in performance as is shown in Figure 5.

## C    ADDITIONAL EXPERIMENTAL RESULTS

### C.1    EXPERIMENTS ON BAIL-TD

We present some of the experimental results for BAIL-TD in this subsection. Figure 4 shows that the performance of BAIL-TD is similar to the performance of naive Behavioral Cloning in Hopper-v2 and Walker2d-v2 environments. The result implies that the BAIL-TD approach is limited in the ability to distinguish between good data points and the bad ones. This is possibly due to the inaccuracy of the trained upper envelope. Recall that in BAIL-TD algorithm, we select state action pairs based on the difference between the values of $r + V(s_i')$ and $V(s_i)$. Since we only use one-step Monte Carlo return, and the value of $r$ is very small compared with the value of $v(s_i)$, the selection is very sensitive to the accuracy of $V(s_i)$.

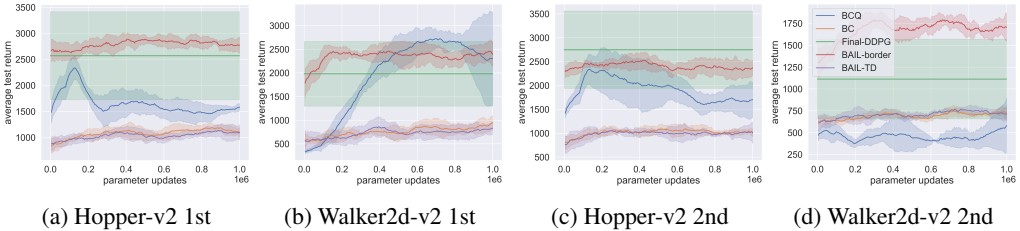

(a) Hopper-v2 1st          (b) Walker2d-v2 1st          (c) Hopper-v2 2nd          (d) Walker2d-v2 2nd

Figure 4: Performance comparison of BAIL-border, BAIL-TD, BCQ, BC and Behavioral Policy (DDPG)

## C.2 Ablation Study for BAIL

We do additional ablation studies for BAIL, where we focus on two heuristic features in the BAIL algorithm: clipping of upper envelope, and return augmentation to approximate non-episodic continuous tasks. We removed each of these features one at a time from the BAIL algorithm and compare with the original BAIL algorithm. Each performance curve is again averaged over 5 independent runs.

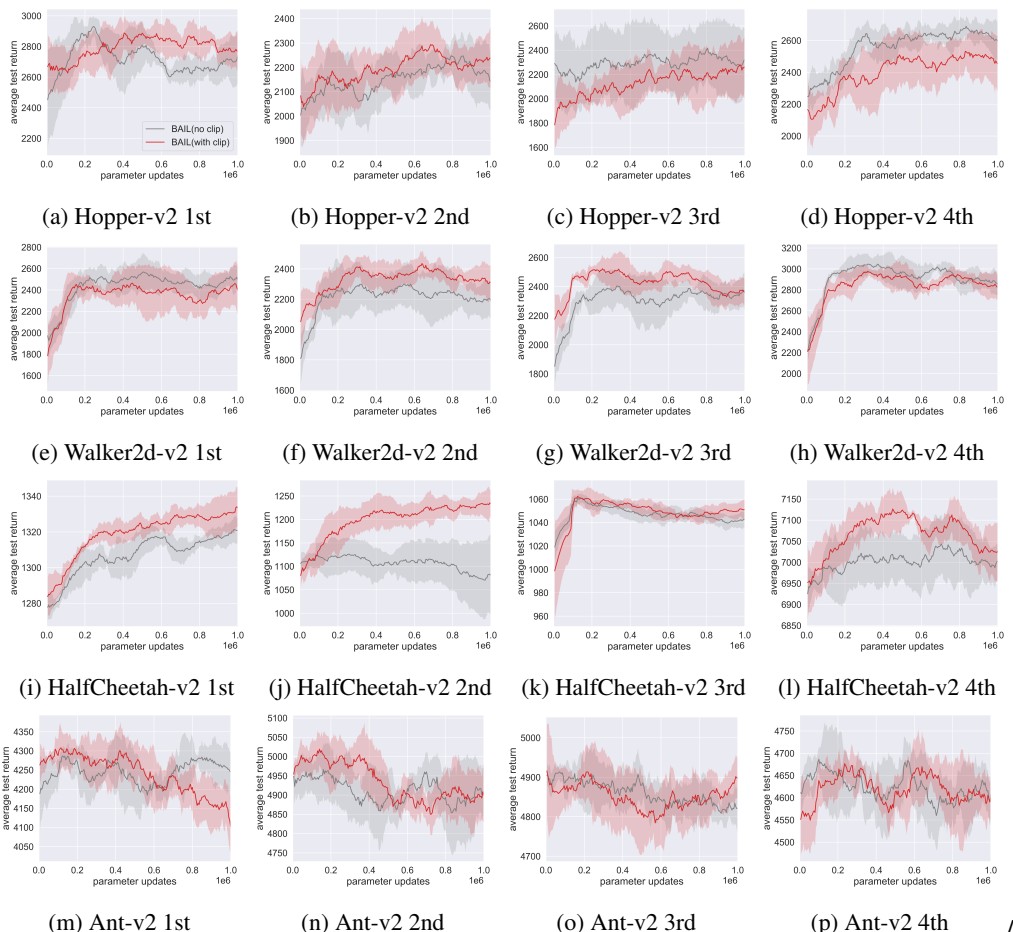

(a) Hopper-v2 1st  (b) Hopper-v2 2nd  (c) Hopper-v2 3rd  (d) Hopper-v2 4th

(e) Walker2d-v2 1st  (f) Walker2d-v2 2nd  (g) Walker2d-v2 3rd  (h) Walker2d-v2 4th

(i) HalfCheetah-v2 1st  (j) HalfCheetah-v2 2nd  (k) HalfCheetah-v2 3rd  (l) HalfCheetah-v2 4th

(m) Ant-v2 1st  (n) Ant-v2 2nd  (o) Ant-v2 3rd  (p) Ant-v2 4th

Figure 5: Ablation of BAIL without Upper Envelope clipping

We found that clipping often, but not always, gives a small boost in performance. As for return augmentation, we found it does not harm the performance of BAIL, and sometimes gives a considerable improvement, particularly for Hopper.

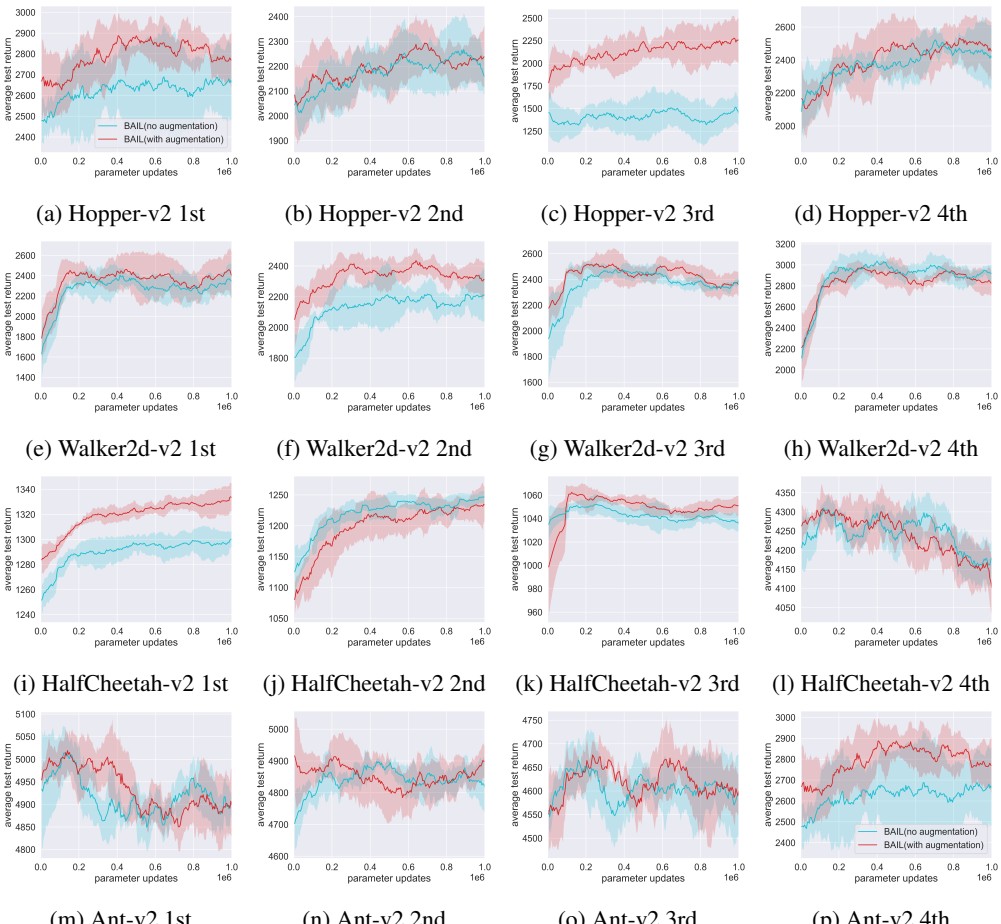

Figure 6: Ablation of BAIL without return augmentation

## C.3    BAIL FOR LOW NOISE-LEVEL DATA

In the main body of the paper we present BAIL in the "Final Buffer" case as described in Fujimoto et al. (2018a), where the exploration noise $\sigma = 0.5$ added to behavior policy is relatively large. In this section we examine the performance of BAIL in a low-noise scenario. To this end, we set $\sigma = 0.1$ and do a similar experiments as were done $\sigma = 0.5$ for the Hopper, Walker and HalfCheetah environments. (Recall that for Ant we used adaptive SAC, which does not have an explicit noise parameter.)

The results are shown in Figure 7. We see that even in this low-noise scenario, BAIL out-performs BCQ by a wide margin for Hopper and Walker, and BAIL continues to out-perform the Final-DDPG policy in most batches. For HalfCheetah, where the Final-DDPG policy gives greatly different results depending on the batch, BAIL's performance is stable and typically near that of the of Final-DDPG policy. After sufficient training, however, BCQ can often do better than the Final-DDPG policy in this environment.

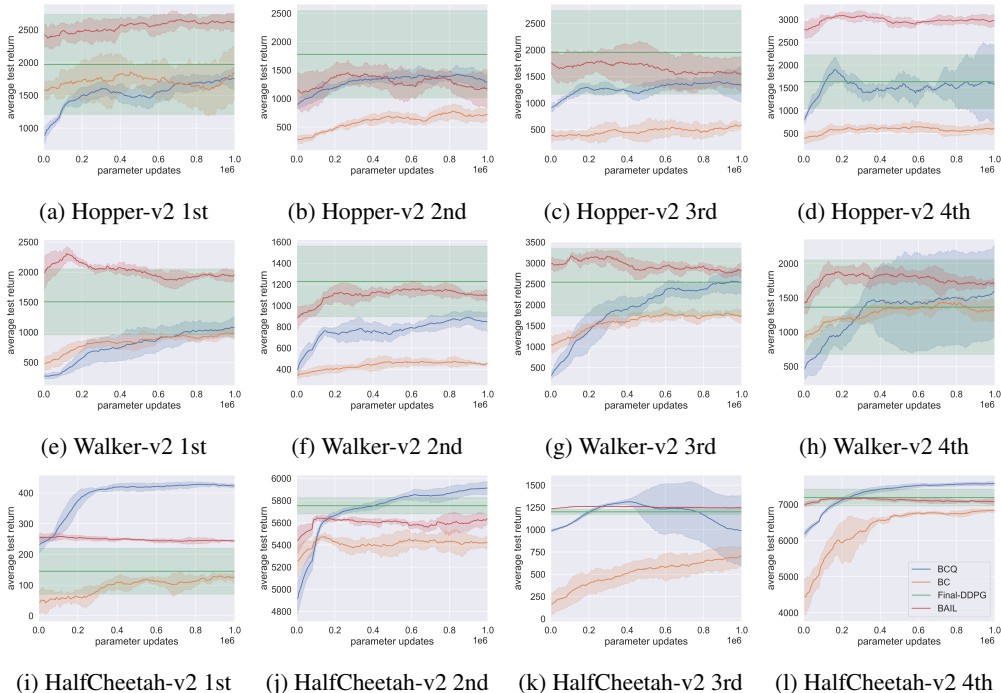

Figure 7: Performance comparison of BAIL, BCQ, BC and Final-DDPG with noise level $\sigma = 0.1$

## C.4 VISUALIZATION OF UPPER ENVELOPES USED

In this section we present the upper envelopes used in the training of BAIL in Figure 2.

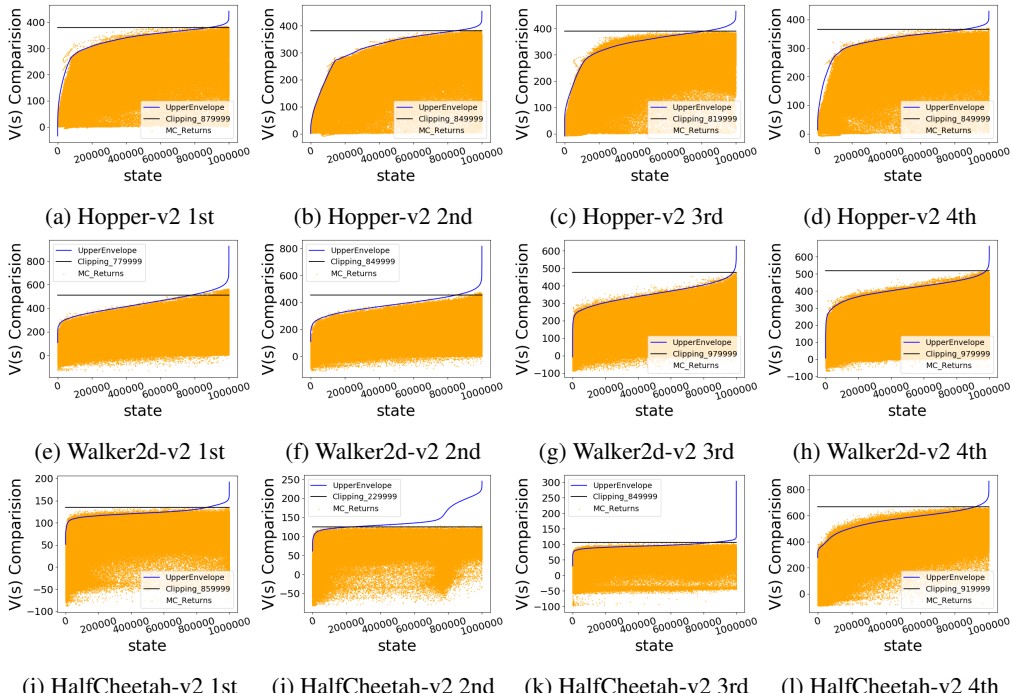

Figure 8: Upper Envelopes of BAIL (DDPG data)

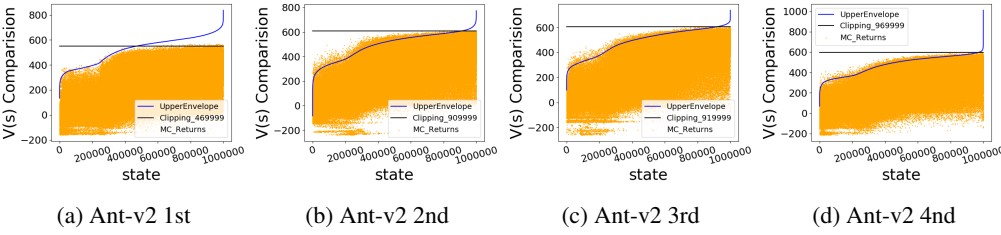

Figure 9: Upper Envelopes of BAIL (SAC data)

