# OpenReview forum: "BAIL: Best-Action Imitation Learning for Batch Deep Reinforcement Learning"
_ICLR.cc/2020/Conference — Reject_

### Official Review · AnonReviewer2 · 2019-10-22
**Official Blind Review #2**

**Rating:** 3

**Review:**

Summary of Claims:

The paper proposes a batch RL method that they claim is simpler than most existing methods that try to avoid the extrapolation error that is prevalent among batch RL methods. They do this by completely avoiding the minimization/maximization (cost/reward) of the approximate value function that is fit to the batch transitions. Instead, they train an approximation for the state value function's tight upper bound (which they refer to as the upper-envelope) by using their monte-carlo returns. By fitting such an approximator, they sample the state-action pairs that are close to the envelope (thus have high values/returns), and use behavioral cloning to fit a parameterized policy to those state-action pairs.

Decision:

Weak Reject.
My decision is influenced by two main reasons:

(1) Although the simplicity of the method is apparent and a very desirable feature, the authors don't highlight situations where this can lead to bad policies. For example, consider that there are two pairs (s, a_1, s') and (s, a_2, s') in the batch that are close to the upper-envelope, and hence will both be used for training the policy. Using Behavioral cloning, the policy would regress to the mean of a_1 and a_2, which could be a terrible action altogether. The issue here is that only one of these two pairs has higher return and our policy needs to only predict that action (or in the case of tie, either one.) This can be really bad in situations where two very different actions can lead to same returns (e.g. in a reacher-like task the arm can reach a goal in two different rotations.) Even though I pointed out a very specific case, one could think of many other cases where the proposed approach might result in a bad policy.

Having said all of this, it might be true that such cases do not appear in practice (which I highly doubt) but its the authors job to raise and clarify that. The current set of experimental setups (mujoco locomotion problems) are not good enough evidence for that and they need experiments where optimal-policies can be multi-modal or have diverse experimental setups (manipulation etc.)

(2) Experimental results are a little unsettling. The primary reason is that in all of the plots, BCQ, BAIL, BC aren't starting from the same test return at 0 parameter updates! In most plots BAIL starts off way higher in return than BCQ, BC with no parameter updates yet, which suggests that the experiments were not setup well. Maybe, they didn't initialize the policy in the same way for all the approaches, maybe the random seeds were not the same for all approaches, or maybe BAIL had some sort of pretraining for the policy that was not accounted for in the parameter updates. In any way, this needs to be addressed. This is also highlighted by the fact that the learning curves for BAIL are almost always flat across a million parameter updates! If you are starting off with a random initialization, there should be an upwards slope for the learning curve. Also, as raised in the previous point I think using these Mujoco locomotion environments is not convincing enough to claim that BAIL is a viable competitive batch RL approach.

Comments and Questions:

(1) I like the simplicity of the approach and the fact that it is much more easier to understand than existing works like BCQ

(2) Paper is well-written. It was clear, lucid and descriptive.

(3) Why is the deterministic dynamics assumption needed? I am curious

(4) The paper makes some subjective statements such as "BEAR is also complex", which is not substantiated well enough. Refrain from making such statements

(5) Not comparing to BEAR because their code is not publicly available is a contentious reason. I personally feel that the authors could have reimplemented it and compared but I am not sure what the community feels about that

(6) Is there any reason why REM cannot be applied to mujoco environments? If it can be, then why did the authors not compare to REM as well?

(7) Another subjective statement (that is clearly wrong) "many robotic tasks are expected to be deterministic environments" - although this is slightly true, the reason we model environments to be stochastic is not because there is inherent randomness in them but because our state descriptions are never complete. The state descriptors are always partial and we account for them by assuming stochasticity in the dynamics. For example, consider a robotic manipulation task where if you know all the environmental factors as part of your state space(such as the friction coefficients) you can assume deterministic dynamics, else you are better off assuming stochastic dynamics because the same actuation might not result in the same motion every time (because of varying friction)

(8) Concatenating subsequent episodes in a batch only makes sense (as the authors point out) if the policy doesn't change much across episodes. But this is not true of current off-policy RL methods like DDPG, SAC. You either need very small learning rate or a trust-region constraint to ensure that the policy doesn't change much across episodes.

(9) Why do different batches with different seeds and the same algorithm lead to widely different results for batch RL? There is clearly something fishy here. Is it because of the off-policy RL methods used to collect the data, is it due to the batch RL method used? More investigation needed

**Experience Assessment:**

I have published one or two papers in this area.

**Review Assessment: Checking Correctness Of Derivations And Theory:**

I carefully checked the derivations and theory.

**Review Assessment: Checking Correctness Of Experiments:**

I carefully checked the experiments.

**Review Assessment: Thoroughness In Paper Reading:**

I read the paper thoroughly.

---

> ### Author Response · Authors · 2019-11-15
> **Response**
>
> Thank you for your thorough review. We appreciate that you "like the simplicity of the approach and the fact that it is much easier to understand than existing works like BCQ".
>
> For your second point you write: "Experimental results are a little unsettling. The primary reason is that in all of the plots, BCQ, BAIL, BC aren't starting from the same test return at 0 parameter updates!" The reason why the BAIL learning curve is  flat is because the imitation learning approach learns very fast. In our experiments, we evaluate the performance of the policy every 5,000 gradient updates, during which 500,000 data points are seen. Therefore, the BAIL policy at the first evaluation point in the plot already is already very good. The experimental results are indeed correct: the simple BAIL algorithm provides better performance than the more complex BCQ algorithm. We have also made our code publicly available so that you try for yourself. The results are not "unsettling".
>
> We agree with your first point, there may be some environments for which BAIL will not do well for reasons you describe (regress to terrible action). But we feel this comment is unfair. Many environments, including all the Mujoco environments, due not have this issue. Note that the BCQ paper also only considers the Mujoco environments, and the BCQ and BEAR could possibly have this problem as well. We feel that our BAIL algorithm should be judged by its performance on existing popular benchmarks and not on imaginary to-be-created benchmarks.
>
> The deterministic assumption is needed for the upper envelope. If the environment is stochastic, then the upper envelope would be enveloping the maximal of the possible returns for a given policy, when the actual value function would be an average of the returns.
>
> We feel that REM and BCQ/BAIL are largely orthogonal. Therefore we feel REM is outside the scope of the paper. In a subsequent paper, we may consider combining BAIL and REM (and doing an ablation study).
>
> In the next version, we will remove the two statements you feel are subjective.
>
> Concerning your last point, nothing fishy whatsoever is going on. This is something we observed that other authors have not brought to light. (Unlike our paper, the BCQ and BEAR papers only looked at one seed to generate a batch. Their results may be different for multiple seeds.) Different batches with different seeds and the same algorithm can indeed lead to widely different results for batch RL. We agree that further work is needed to understand this phenomenon. But we also feel that is question should be addressed in separate paper.
>
> Given our response above and the veracity of our experimental results, would you consider raising your score? Don't you feel that the novelty, simplicity, and performance of the algorithm on the existing competing benchmarks should be the main criteria for scoring?

---

### Official Review · AnonReviewer3 · 2019-10-23
**Official Blind Review #3**

**Rating:** 1

**Review:**


The paper tries to solve a batch reinforcement learning problem with a very simple but efficient algorithm. It first learns a smooth upper bound of Monte Carlo returns in the batch data (called the "upper envelope"). Then, the algorithm chooses state action pairs of the batch data that have returns larger than constant times the upper envelope. It lowers the constant until the algorithm gets 25% of the data. Then the algorithm trains the policy on chosen state-action pairs. The algorithm is shown to outperform BCQ in experiments.

Although I like the idea of the paper, I vote for rejection. While there is no theoretical guarantee on the performance of the algorithm, the design of the algorithm does not follow the usual design the other researchers follow. The way of reporting the experiment results does not seem very professional. I recommend the authors to consult with some other researchers who have publication experience. In the current form, the paper is very poor in detail that makes readers hard to be convinced with the results.

These are some points that I could not understand:

1. Why do you fix K=10000 on modified loss instead of dual gradient descent for constrained optimization?
2. How do you guarantee that choosing (s,a) such that G>xV gives you good samples? Since mean returns are not zero, it won't pick the top 25% actions for all states. States with the high mean return will have all of its samples included, while states with the low mean return will have all of its samples excluded. Although the authors concatenated all the experiences to compute returns (which is ad-hoc as well), the initial states will have a lower return than other states. This means that most of the actions of the initial states will be excluded in the training set while more actions of the other states will be included, which does not seem desirable. (e.g. in Figure 1 Ant. If we set x=0 (extreme case), states of timestep >600000 will be all included where t<600000 will be partially excluded. )
3. In the explanation of Figure 2, it is written as "standard deviation confidence interval". Is it standard deviation, or confidence interval? Also, why are the standard deviation in the Figure 2 and the Table 1 so different? How do you compute Improvement in the Table 1? What happens if the environment gives negative returns only (i.e. Pendulum), such that BCQ gives you -1000 and BAIL gives you -500?
4. As claimed in theorems, V=max(G) if lambda->infinity. This means that the "Highest Returns" in figure 3 is also one specific hyperparameter choice of the suggested algorithm. There might be a better choice of regularization that outperforms both BAIL and Highest Returns as early-stopping done in the paper is just one random amount of regularization. What was the early-stopping criterion and how is it chosen? How do we know it is the best regularization option?
5. Is the final DDPG or final SAC evaluated with a deterministic policy? According to the paper, I assume that it was not. Those algorithms usually add large noise while training for exploration, and such noise is removed while in evaluation. In Bear Q learning, better action selection technic is used, which chooses the action sample that maximizes critic Q. Is the evaluations really fair for all algorithms? As far as I know, Mujoco environments are deterministic except the initial state sampling, and there should only be very small variance.

Also, I believe the paper should be compared to Bear Q learning as well, as it is very easy to implement and outperforms BCQ by a large margin.


**Experience Assessment:**

I have read many papers in this area.

**Review Assessment: Checking Correctness Of Derivations And Theory:**

I carefully checked the derivations and theory.

**Review Assessment: Checking Correctness Of Experiments:**

I carefully checked the experiments.

**Review Assessment: Thoroughness In Paper Reading:**

I read the paper at least twice and used my best judgement in assessing the paper.

---

> ### Author Response · Authors · 2019-11-15
> **Response to reviewer 1**
>
> Thank you thorough review. Below we respond to many of your comments.
>
> We feel the paper should largely be judged on the novelty, simplicity, and efficacy of the BAIL algorithm. The BAIL algorithm is significantly simpler than BCQ but nevertheless provides better performance. These contributions alone merit acceptance at ICLR.
>
> You write "the paper is very poor in detail that makes readers hard to be convinced with the results." Nevertheless, Reviewer 2 writes "Paper is well-written. It was clear, lucid and descriptive." We can assure you that there are some senior authors on the paper. The experimental results are correct, as we discuss below.
>
> 1. Whether we use dual-gradient descent or a penalty with K =10,000 to solve a constrained optimization problem is a matter of taste. The authors have significant experience in constrained optimization, and prefer the penalty approach for this problem. The K value here is a hyper-parameter. What is most important, however, is that approach as is works: using a simple loss function, it beats BCQ by a wide margin. We feel you should give more credit to the simplicity and novelty of the algorithm, and to the fact that it provides excellent performance.
>
> 2. Your point (2) is well-taken. However, like most of deep learning, our BAIL algorithm is a heuristic that is motivated by mathematics but does not have a mathematical guarantee. What is most important is that it is simple and it works, as shown in the experimental results.
>
> 3. It is a one-standard deviation confidence interval. We will clarify this in the next version. We will also clarify how we compute the improvement.
>
> 4. For early stopping, we took the standard approach of monitoring the loss function over a validation set (which is chosen from the batch data). In the next version, we will clarify this.
>
> 5. Yes, all the final polices are evaluated with a deterministic policy, not with a stochastic policy.
>
> Also, we feel your last comment is unfair. At the time of submission, the BEAR code was not available. We feel that BEAR is complex (as compared to BAIL) and would have been difficult to implement. Furthermore, even if we had implemented it, reviewers may have then questioned our implementation. It is the responsibility of the authors of the BEAR paper to provide their code, and not ours to figure out how it should be implemented. It is our understanding that the BEAR code has been recently released. However, this submission should be judged in the context of what code was publicly available at the time of release.
>
> You seem to be surprised that the experimental results can be so good with such a simple algorithm. But the fact is that it's true. One doesn't need VAE's and other machinery to achieve good results. We made our code available at the time of submission, and the reviewers are invited to check it out for themselves.

---

### Official Review · AnonReviewer4 · 2019-11-02
**Official Blind Review #4**

**Rating:** 3

**Review:**

Summary:
This paper studies the problem of learning a policy from a fixed dataset. The authors propose to estimate a smooth upper envelope of the episodic returns from the dataset as a state-value function. The policy is then learned by imitating the state action pairs from the dataset whose actual episodic return is close to the estimated envelope.

Recommended decision:
The direction of imitating "good" actions from the dataset is interesting. The intuition of estimating an upper envelope of the value function seems reasonable. However, I feel like this paper is not ready to be published in terms of its overall quality, mainly due to the lack of correctness, rigorousness and justification in statements and approaches.

Major comments:

- On the top of page 4:  "Because the Mujoco environments are continuing tasks, it is desirable to approximate the return over the infinite horizon, particularly for i values that are close to the (artificial) end of an episode. To do this, we note that the data-generation policy from one episode to the next typically changes slowly. We therefore apply a simple augmentation heuristic of concatenating the subsequent episode to the current episode, and running the sum in (1) to infinity." I cannot see how this approach is validated. The reset of initial state makes cross-episode cumulative reward from a state s not an approximation to the real return from state s. Estimating the infinite horizon return from finite horizon data is indeed a challenge here and simply cut the return at the end of an episode is be problematic. But the solution proposed by the authors is wrong in principle and cannot be simply justified by "good empirical performance". I feel hard to regard this choice a valid part of an algorithm unless further justification can be provided.

- Statements of theorems (4.1 and 4.2) are non-rigorous and contain irrelevant information: "lambda-smooth" is not an appropriate terminology when lambda is the weight of the regularizer. The actual "smoothness" also depends on the other term in the loss (same lambda does not indicate same smoothness in different objectives). For the same reason, Theorem 4.2 is wrong as changing K also changes the smoothness of the learned function. Proof of Theorem 4.2 in appendix is wrong as the authors ignore the coefficients in the last equation. Theorem 4.1-(1) cannot be true unless how V_\phi is parameterized is given: e.g. if there is no bias term or the regularization is applies to the bias term V will always output 0 as lambda 0-> \infty. The "2m+d" in Theorem 4.1-(2) is irrelevant to this work and cannot be justified without more detailed statements about how the network is parameterized. I appreciate the motivation that the authors try to validate the use of their objective to learn a "smooth upper envelope" but most of these statements are somewhat trivial and/or wrong section 4.1 does not actually deliver a valid justification.

- The use of "smooth upper envelope" itself can bring both over-estimation and under-estimation. For example, if one can concatenate different parts from different episodes to get a trajectory with higher return, the episodic return for the states along this trajectory is an under-estimate. Although it is fine to use a conservative estimate it would be better to be explicit about this and explain why this may not be a concern. On the other hand, it can bring over estimation to the state-values due to the smoothness enhanced to the fitted V. It would be better to see e.g. when these concerns do not matter (theoretically) or they are not real concerns in practice (by further inspecting the experiments).

- Regarding Experiments: Why Hopper, Walker, HalfCheetah are trained with DDPG while Ant is trained by SAC? The performance of Final-DDPG/SAC after training for 1m steps looks way below what SAC and TD3 can get. Is it because they are just partially trained or noise is added to them? The baseline online-trained policy should not contain noise for a fair comparison. That said, in batch RL setting it is not necessary to compare to online-trained policy because it is a different setting. But if the authors want to compare to those, choice of baseline should be careful. An important baseline which is missing is to run vanilla DDPG/TD3/SAC as a batch-mode algorithm.


Minor comments:

- Section 3, first paragraph: It is not very meaningful to say "simulators are deterministic so deterministic environments are important". Simulators are made by humans so they can be either deterministic or stochastic. "many robotic tasks are expected to be deterministic environments" is probably not true. I do not view "assuming deterministic envs" as a major limitation but I do not find these statements convincing as well. Similarly, the argument for studying non-stationary policy seems unsupportive: if the dataset comes from training a policy online then why do we care about learning another offline policy rather than just use or continue training the online policy. One argument I can see is that the online policy is worse. But the fact that these policies are worst than running e.g. SAC for a million steps makes the motivation questionable. Again, I do not view "choice of setting" as a limitation but I just find these statements a bit unsupportive.


Potential directions for improvement:

To me the main part of the paper that looks problematic is Section 4.1 (both the approximation of infinite horizon returns and the theorems). It would be better to see a more rigorous and coherent justification of this approach (or some improved version), e.g. by either presenting analysis that is rigorous, correct and actually relevant or leave the space for more detailed empirical justification (e.g. whether potential over/under-estimating happens or not, comparing the estimated V to real episodic return of the learned policy).


**Experience Assessment:**

I have published one or two papers in this area.

**Review Assessment: Checking Correctness Of Derivations And Theory:**

I carefully checked the derivations and theory.

**Review Assessment: Checking Correctness Of Experiments:**

I assessed the sensibility of the experiments.

**Review Assessment: Thoroughness In Paper Reading:**

I read the paper at least twice and used my best judgement in assessing the paper.

---

> ### Author Response · Authors · 2019-11-15
> **Response to your review**
>
> Thank you for your thoughtful review. As you and the other reviewers have noted, the approach of imitating good actions is novel as so is the upper envelope. The principal contributions of the paper are(1) introduce these novel approaches, and (2) through careful and thorough evaluation show that BAIL soundly beats BCQ, thereby achieving state of the art performance. We feel the paper should be largely judged by these two contributions.
> Your concerns about the rigor of the theorem statements seems to be mostly a semantic one and can easily be corrected. Given the high novelty (and simplicity) of the approach, and the excellent experimental results, perhaps you can consider raising your score for this paper? If you like, we can remove the "smoothing" terminology and/or remove the theorems altogether from the paper.
>
> The BCQ paper considers only the environments Hopper, HalfCheetah, and Walker, and uses batch data sets generated by DDPG. In order to provide a fair comparison with the BCQ paper, we choose the same environments and batch generating techniques. We also wanted to enlarge the scope of the experimentation by including Ant. However, DDPG for Ant fails to learn and generate useful batch data. For that reason, we used SAC to generate the batch data for Ant.

---

### Decision · Program_Chairs · 2019-12-19

**Decision:**

Reject

**Comment:**

The authors propose a novel algorithm for batch RL with offline data. The method is simple and outperforms a recently proposed algorithm, BCQ, on Mujoco benchmark tasks.

The main points that have not been addressed after the author rebuttal are:
* Lack of rigor and incorrectness of theoretical statements. Furthermore, there is little analysis of the method beyond the performance results.
* Non-standard assumptions/choices in the algorithm without justification (e.g., concatenating episodes).
* Numerous sloppy statements / assumptions that are not justified.
* No comparison to BEAR, making it challenging to evaluate their state-of-the-art claims.
The reviewers also point out several limitations of the proposed method. Adding a brief discussion of these limitations would strengthen the paper.

The method is interesting and simple, so I believe that the paper has the potential to be a strong submission if the authors incorporate the reviewers suggestions in a future submission. However, at this time, the paper falls below the acceptance bar.